# Identifying Cancer Stage-Related Biomarkers for Lung Adenocarcinoma by Integrating Both Node and Edge Features

**DOI:** 10.3390/genes16030261

**Published:** 2025-02-24

**Authors:** Zige Wang, Hamza Benhammouda, Bolin Chen

**Affiliations:** 1School of Basic Medicine, Shaanxi University of Chinese Medicine, Xianyang 712046, China; 223010512690@email.sntcm.edu.cn; 2School of Computer Science, Northwestern Polytechnical University, Xi’an 710072, China; ayoub.fouad995@gmail.com

**Keywords:** edge biomarker, feature selection, precision oncology, cancer stage

## Abstract

**Background**: In order to characterize phenotypes and diseases, genetic factors and their interactions in biological systems must be considered. Although genes or node features are the core units of genetic information, their connections, also known as edge features, are composed of a network of gene interactions. These components are crucial for understanding the molecular basis of disease and phenotype development. Existing research typically utilizes node biomarkers composed of individual genes or proteins for the binary classification of cancer. However, due to significant heterogeneity among patients, these methods cannot adapt to the subtle changes required for precise cancer staging, and relying solely on node biomarkers often leads to poor accuracy in classifying cancer staging. **Methods**: In this study, a computational framework was developed to diagnose lung adenocarcinoma, integrating node and edge features such as correlation, covariance, and residuals. The proposed method allows for precise diagnosis in the case of a single sample, which can identify the minimum feature set that effectively distinguishes cancer staging. **Results**: The advantages of the proposed method are: (i) it can diagnose each individual test sample, promoting personalized treatment; (ii) integrating node and edge features can improve diagnostic accuracy, indicating that each type of feature can capture unique aspects of the disease; (iii) it significantly reduces the number of features required to accurately classify the four stages of cancer, thereby achieving optimal cross-validation accuracy. **Conclusions**: This streamlined and effective feature set highlights the potential of our approach in advancing personalized medicine and improving clinical outcomes for cancer patients.

## 1. Introduction

Complex diseases such as lung adenocarcinoma (LUAD) require a deep understanding of molecular dynamics, extending from individual molecular components to interactions within biological networks. This viewpoint is supported by the concept of ’edgetics’ in online medicine, which emphasizes the importance of gene interactions in disease mechanisms [1,2,3,4,5,6]. In this study, we introduce a new computational framework aimed at diagnosing LUAD for individual samples, by integrating node features (genes) and edge features (gene interactions), such as correlation, covariance, and residuals.

The traditional methods for characterizing these features typically require the characteristics of node biomarkers, ranking the expression levels of individual molecules without considering their interactions [7,8,9,10]. M. Mairé et al. [7] used 54 gene expression signatures as biomarkers for predicting lymph node metastasis, providing a promising direction for early staging of cancer. However, the study’s limitations of small sample size and single-center sample size have raised concerns about the broader applicability of its research findings. M. Heck et al. [8] discussed the molecular lymph node staging of bladder cancer, revealing how specific biomarkers can improve patient management by more accurately predicting the risk of lymph node metastasis and recurrence. T. Osako et al. [9] developed a new molecular based lymph node staging system that is significantly correlated with patient prognosis, highlighting the crucial role of precise biomarker-driven staging in treatment strategies. G. E. Lind et al. [10] studied the epigenetic biomarkers of colon cancer and proposed their prognostic relevance in sentinel lymph nodes. Despite these innovative methods, the challenge remains in creating biomarker panels that consistently produce substantial results across different patient populations [11].

In recent years, edge biomarkers have become a key development in cancer staging, aimed at overcoming the inherent limitations of traditional node biomarker methods [12,13,14,15]. This shift from observing genes in isolation to analyzing their interactions helps to gain a deeper understanding of the complex molecular dynamics in cancer. Lu et al. [12] emphasized the practicality of edge biomarkers and developed a dynamic edge-based biomarker that can non-invasively predict hepatocellular carcinoma (HCC) in hepatitis B patients through blood testing. N. Adnan et al. [13] discussed the robustness of marginal biomarkers in predicting breast cancer metastasis. Their research indicates that analyzing the interaction data of gene pairs provides a more reliable predictive model than traditional single-gene analysis, improves the accuracy of metastasis prediction, and highlights the potential of these biomarkers in clinical applications. W. Zhang et al. [14] further developed the concept of the “EdgeBiomarker” framework, which aims to diagnose the phenotype of cancer at different stages by analyzing edge biomarkers. J. A. Ludwig et al. [15] have consistently supported the integration of edge biomarkers, documenting significant advances in genomics and molecular pathology that support the use of edge biomarkers.

Although progress has been made in utilizing edge biomarkers, most studies have used the Pearson correlation coefficient (PCC) to measure gene interactions [16,17,18,19,20]. This exclusive dependence on PCC not only limits the understanding to a small part of gene interaction dynamics, but also tends to overlook other important types of interactions which can provide deeper insights into molecular dynamics and are crucial for the detailed staging of cancer. The attention to PCC in the literature suggests an important opportunity to expand the scope of methods to capture additional dimensions of gene interactions, paving the way for more comprehensive and effective cancer staging strategies.

In this study, a computational framework was proposed that innovatively integrates node and edge features to enhance the classification of lung adenocarcinoma staging. To identify important node features, the EdgeR [21] tool was first used to determine a list of differentially expressed genes (DEGs) (the tool can be found at https://bioconductor.org/packages/release/bioc/html/edgeR.html, accessed on 11 November 2024). Then, a strict false discovery rate (FDR) was used to ensure that only the most significantly altered genes were considered, and a multiclass classification using random forest classifier was employed to quantify the relationship between lymph node features and lung adenocarcinoma staging. Gene interaction dynamics were quantified using the Pearson correlation coefficient (PCC), covariance, and residuals to capture complex interactions between gene expression under normal and disease conditions. Each method provided a unique perspective on gene interactions and identified significant deviations from normal values, indicating potential biomarkers for LUAD. Then, these features were sorted and the most predictive features were selected as the comprehensive feature set, which is expected to enhance our understanding and prediction of cancer stages.

The proposed method was validated on a lung cancer dataset. It can successfully identify the smallest feature set that can distinguish cancer staging [22]. This achievement not only highlights the potential of our method in simplifying the diagnostic process, but also highlights its contribution to promoting personalized medicine for cancer patients and improving clinical outcomes.

## 2. Materials and Methods

### 2.1. Data Collection and Preprocessing

The RNA-seq data related to lung adenocarcinoma (LUAD) were obtained from the Cancer Genome Atlas (TCGA) [23] and UCSC Xena repository [24]. The dataset consists of a feature matrix representing gene expression profiles and the corresponding phenotype data indicating cancer staging. In this study, only samples containing stage information were retained, and the sample distribution is as follows: normal (59 samples), stage I (266 samples), stage II (120 samples), stage III (84 samples), and stage IV (26 samples).

The preprocessing of the LUAD dataset began with the normalization of gene expression values, utilizing logarithmic transformation to standardize the data size and improve the comparability of patient samples. This dataset included gene expression profiles of 31,823 transcripts and required strict quality control measures. Transcripts with zero expression in over 50% of the samples were excluded to eliminate non-informative variables and enhance the analytical robustness of the study. The final number of genes retained for analysis in this study was 13,163.

### 2.2. Node Feature Generation

Differential expression analysis was first performed using EdgeR (Version 4.4.2) [21], a statistical software package tailored for RNA-Seq data that can adapt to sample specific changes and biological complexity. The generalized linear model (GLM) framework of EdgeR was used to compare and analyze the different stages of LUAD, and a differential expression spectrum was established. This method can quantify folding changes and statistical significance, and identify differentially expressed genes (DEGs) at different stages of cancer. The selection of DEGs is strictly controlled by statistical thresholds, particularly the false discovery rate (FDR), to ensure the detection of genes with significant expression changes.

In order to focus on genes that consistently change at all stages, these lists were crossed over rather than combined. The basic principle of using intersection instead of union in this case is to separate genes that are consistently differentially expressed at all stages, which may play a key role in the progression and pathology of LUAD at each stage. This can reduce noise and improve the credibility of results by focusing on genes that exhibit consistent changes, rather than capturing any genes that exhibit variations at any stage.

A 10-fold cross validation method was used in this study. Once the training and testing sets were ready, a random forest classifier was used to elucidate the importance of various features in predicting disease stages. The random forest algorithm is renowned for its robustness and versatility, and is a powerful tool in our analysis. By utilizing a set of decision trees, classifiers can effectively capture complex relationships in data and make accurate predictions. Therefore, we calculated feature importance based on the trained classifier.

### 2.3. Analyzing Gene Interaction Dynamics

#### 2.3.1. The Role of Correlation in Capturing Gene Interactions

The Pearson correlation coefficient (PCC) is a widely used statistical measure used to evaluate the linear relationship between two continuous variables. In the context of genomic research, Pearson correlation is particularly valuable for exploring the interactions between genes, providing insights into how gene expression covaries across different samples or conditions.

The *PCC* is mathematically defined as:$$PCC_{xy}=\frac{\sum \left(x_{i}-\bar{x}\right)\left(y_{i}-\bar{y}\right)}{\sqrt{\sum {\left(x_{i}-\bar{x}\right)}^{2}{\left(y_{i}-\bar{y}\right)}^{2}}}$$
where $PCC_{xy}$ represents the Pearson correlation coefficient between variables x and y, $x_{i}$ and $y_{i}$ denote individual data points, and $\bar{x}$ and y denote the means, respectively.

Pairwise gene–gene Pearson correlation coefficients of n normal samples are denoted as $PCC_{n}$, which indicates the interaction relationship between the two genes in the normal sample, and the value has strength and direction. Once a patient sample is added, the PCC value of those n+1 samples become $PCC_{n+1}$, where the difference can be represented as [25]$$\Delta PCC=PCC_{n+1}-PCC_{n}$$
which highlights the changes in correlation introduced by individual diseased samples. The value of ΔPCC for individual gene pairs can be used as edge features, and a similar random forest algorithm can be used to identify the top *PCC*-related edge biomarkers together with the experiment of cross validation.

#### 2.3.2. The Role of Covariance in Capturing Gene Interactions

Covariance provides a method to measure how changes in one variable are correlated with changes in another variable, which is highly valuable in genomics for examining relationships between gene expression. This statistical data is particularly important for understanding how gene activity changes together under different biological states or conditions.

The calculation of covariance involves determining the average product of the deviations between each pair of variables (gene expression) and their respective means. Mathematically, it is defined as:$$COV\left(X,Y\right)=E\left[\left(X-\mu_{X}\right)\left(Y-\mu_{Y}\right)\right]$$
where X and Y are two random variables representing gene expression levels, and $\mu_{X}$ and $\mu_{Y}$ are their means.

Unlike Pearson correlation, which normalizes covariance by multiplying the standard deviations of two variables, this method provides a dimensionless coefficient that preserves the scale of the original data. This feature is particularly useful when the magnitude of the changes in expression has biological significance.

The difference in covariance, expressed as ΔCOV, can be calculated as follows:$$\Delta COV=COV_{n+1}-COV_{n}$$
which determines the degree of change in the disease state sample from its baseline to the new state. Figure 1 shows the workflow for identifying the top 20 covariance edge features, which is very similar to generating Pearson correlation edge features.

#### 2.3.3. The Role of Residual Errors in Capturing Gene Interactions

In gene expression analysis, residuals represent the difference between observed values and predicted values from statistical models (usually linear regression models). After considering the influence of other variables in the model, these errors capture the unexplained variability in gene expression levels.

The theory behind residuals originates from the principle of linear regression analysis. In linear regression, the relationship between variables is modeled using straight lines, and residuals represent the vertical distance between observed data points and the regression line. These errors reflect to what extent the model is unable to explain changes in the data, providing clues for other factors that affect gene expression beyond the variables included in the model.

For the normal stage group, linear regression is used to describe the relationship between the expression levels of each pair of genes. Each regression line is described by its slope, which represents the change in response of one gene expression to another gene and reflects the strength of their interaction. The y-intercept is the baseline expression level of one gene at the lowest activity of another gene. Therefore, it provides the starting position for interaction. Figure 2 shows the workflow for identifying the top 20 residual edge features, where the method was adopted from [26].

Residuals provide valuable information about gene interactions, which traditional correlation-based methods such as Pearson correlation coefficient or covariance may not be able to capture. Although PCC and covariance focus on capturing linear relationships between gene pairs, residuals reveal deviations from these linear trends, highlighting nonlinear interactions and regulatory complexity within biological systems.

### 2.4. Integrated Feature for Cancer Stage Classification

In this study, the optimization of an integrated feature matrix involves a systematic feature selection and integration method to improve the accuracy of cancer staging classification. This section provides an overview of the theoretical framework and methods used to implement this optimization, without delving into the detailed experimental steps.

The proposed method involves identifying and selecting the top 20 features from each of the four key feature types: node attributes, Pearson correlation coefficient, regression residuals, and covariance measures. Selecting from this set of features allows us to capture various aspects of biological data and provide a comprehensive representation of potential molecular interactions.

In order to optimize the integrated feature matrix, a new feature selection strategy was adopted, aimed at maximizing the F1 score while controlling the total number of features. The process started with evaluating feature combinations, starting with at least one feature from each type and gradually increasing the total number of features. This incremental method allowed for the monitoring of changes in the F1 score with each addition. It is worth mentioning that the F1 score used in this study was a multi-class F1 score, since the classifier we used was also a multi-class classifier.

Figure 3 illustrates how different feature types are integrated and identifies the most important feature combination set for cancer staging classification, providing a visual representation of the method and results.

## 3. Results

### 3.1. Performance of the Top 20 Node Features

After differential expression analysis, the number of identified DEGs was 407 in stage I, 402 in stage II, 373 in stage III, and 143 in stage IV. This crossover method generated a comprehensive list of 143 DEGs present in all stages of cancer.

After quantifying the importance of features, the obtained features can be evaluated using a recursive feature elimination with cross validation (RFECV) procedure, which systematically prunes features based on their importance and iteratively eliminates features with low correlation when evaluating model performance. Since the goal of this study was to identify the smallest but representative biomarkers, the top 20 node features were selected as node biomarkers. This choice was based on professional knowledge in the field, considering that 10 features may be too limited, while 30 features may be too many for clinical practice. It can also be seen in Figure 4 when evaluating the relationship between F1 score and the number of node features, which indicates that the performance of the model does not significantly improve beyond 20 features.

Table 1 summarizes the detailed performance of the random forest classifier and the top 20 most important node features. These visualizations and tabular representations provide a clear empirical basis for the proposed method selection.

### 3.2. Performance of the Top 20 Edge Features

To ensure the reliability and generalizability of the proposed model, a 10-fold cross validation method was used, followed by edge feature segmentation using the random forest algorithm. Figure 5 shows the relationship between F1 score and the number of PCC edge features. After an in-depth evaluation, and based on expert opinions in the field, we decided to use the top 20 features determined by RFECV as our final model.

Table 2 shows the top 20 Pearson correlation edge features obtained in this experiment.

Table 3 and Table 4 show the top 20 covariance correlations and residual edge features obtained during the experiment.

### 3.3. Optimal Feature Count Determination

To determine the optimal number of features for our integrated feature matrix, which included individual genes, changes in Pearson correlation coefficients (PCC), covariance, and regression residuals, we employed recursive feature elimination (RFE) and combined it with cross validation (RFECV). This method systematically removes features, constructs models, and evaluates their performance at each step.

As the number of features increase, the model performance initially shows improvement. This trend is shown in Figure 6. The relationship between the average F1 score and the total number of features indicates that as more features are added, the F1 score will increase, with a peak around 0.98. However, after this peak, performance begins to decline, indicating that further additions generate more noise than information values, thereby reducing the efficiency of the model.

Using the grid search method combined with cross validation, we systematically explored different feature combinations. The structure of this search included different types of features: node attributes, Pearson correlation coefficient, residuals, and covariance measures. We evaluated each combination to determine its effectiveness in improving model performance.

The results of the grid search showed that the best performing feature combination was the balanced combination of all considered types. This optimal set included one node attribute, four Pearson correlation coefficients, two residuals, and four covariance measures. Table 5 summarizes the detailed information.

Table 6 provides a detailed list of the results of this comparative analysis, revealing important insights. The comprehensive feature combination utilized the combination of all feature types and achieved a significantly higher F1 score of 0.982. Although models using only one feature (single gene, PCC, covariance, or residual) produced observable F1 scores ranging from 0.912 to 0.955, they did not match the performance of the ensemble method.

The outstanding performance of the integrated feature set highlights the synergistic effect of combining different types of functions. This synergistic effect enables the model to capture a wider and more complex range of information features, significantly improving the accuracy of classification.

### 3.4. Comparison with Other Methods

To further evaluate the effectiveness of our integrated feature selection strategy, a comparative experiment was conducted, which included not only the proposed method but also several established methods. The purpose was to compare the performance of the proposed method with traditional methods and verify its superiority in identifying biomarkers related to staging of lung adenocarcinoma (LUAD).

Table 7 shows the F1 score and the number of features used for each method, including the new integrated approach.

Comparative analysis shows that WGCNA achieved an F1 score of 0.403 out of 151 features, while ReliefF achieved an F1 score of 0.422 out of 100 features. Our previous research method SFR-GSN achieved an F1 score of 0.968 in 7 features. In contrast, our new method integrated 11 features from different types (single gene, PCC, covariance, and residuals) and achieved an excellent F1 score of 0.982.

This comprehensive comparison highlights the advantages of our integrated approach in determining highly specific and clinically relevant biomarkers for staging lung adenocarcinoma, paving the way for its potential applications in clinical diagnosis and personalized medicine.

## 4. Discussion

To verify the effectiveness of the identified biomarkers, Kyoto encyclopedia of genes and genomes (KEGG) pathway analysis and gene ontology (GO) enrichment analysis were conducted. After selecting 11 optimal instrument features for staging lung adenocarcinoma (LUAD), a series of non-overlapping gene IDs were extracted from these selected features. We identified genes related to edge features from each of the 11 selected features, resulting in a list of 19 unique genes.

Significant tests were conducted on these 19 genes in the biomarkers of LUAD stages, resulting in 12 genes being enriched in 10 pathways. Among them, the RHEB gene is involved in six pathway processes and is strongly related to lung cancer [27,28]. In Figure 7, red dots represent genes, while lines of different colors represent various pathways. Each endpoint of a line represents a gene, while the other endpoint represents the center of the pathway. The size of the center is proportional to the number of enriched genes in the pathway.

The x-axis of the figure displays the enrichment score, which quantifies how these processes are enriched, while the y-axis lists various biological processes. Each color in the figure represents one of the three main categories of GO, among which only the top 10 terms ranked by richness score are displayed.

These analyses confirm the biological relevance of the selected features, where key genes are involved in multiple pathways and biological processes, enhancing the potential of comprehensive feature sets for precise cancer staging classification and their applicability in clinical diagnosis and personalized medicine.

Gene ontology (GO) enrichment analysis was also conducted, and the results are shown in Figure 8.

For future studies, there are still some challenges that have not been well addressed as follows. Firstly, the effectiveness of the model largely depends on the quality and completeness of the genomic dataset. Any inconsistency in data collection or processing can significantly affect the performance of the model. In addition, the initial stages of feature selection and training are resource intensive and there is a risk of overfitting. Addressing these issues requires rigorous validation techniques and careful management of data quality and model complexity to ensure robustness and universality in different clinical environments.

## 5. Conclusions

In this study, a new method for analyzing biomarkers related to lung adenocarcinoma staging was introduced, which combines lymph node and edge features. This method is an extension of traditional methods, in which only single node features are used to generate diagnostic biomarkers. Here, edge features are based on correlation, covariance, and/or residuals.

Numerical experiments on cancer datasets have shown that this method can effectively distinguish cancer staging by using the smallest edge feature set, and is significantly superior to existing methods. The efficiency of the proposed method provides potential capabilities for promoting personalized medicine for cancer patients and improving clinical outcomes.

## Figures and Tables

**Figure 1 genes-16-00261-f001:**
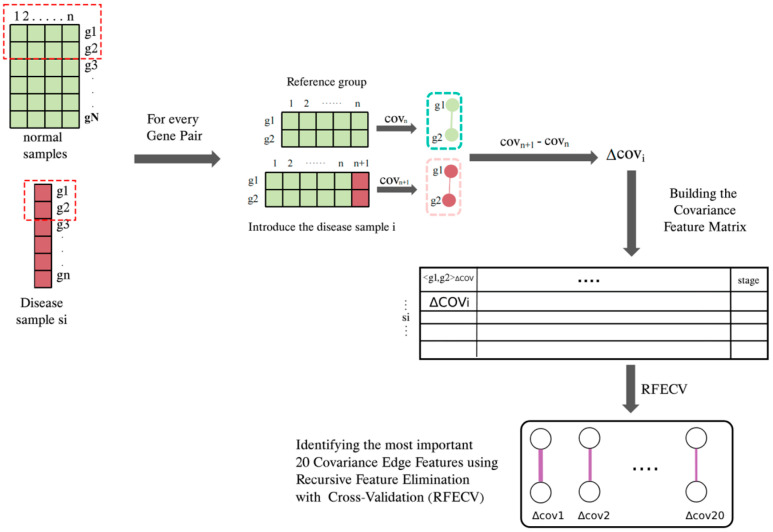
Workflow of identifying the top 20 covariance edge features. The number on the top red box is the samle IDs, and the symbol on the right are the gene IDs.

**Figure 2 genes-16-00261-f002:**
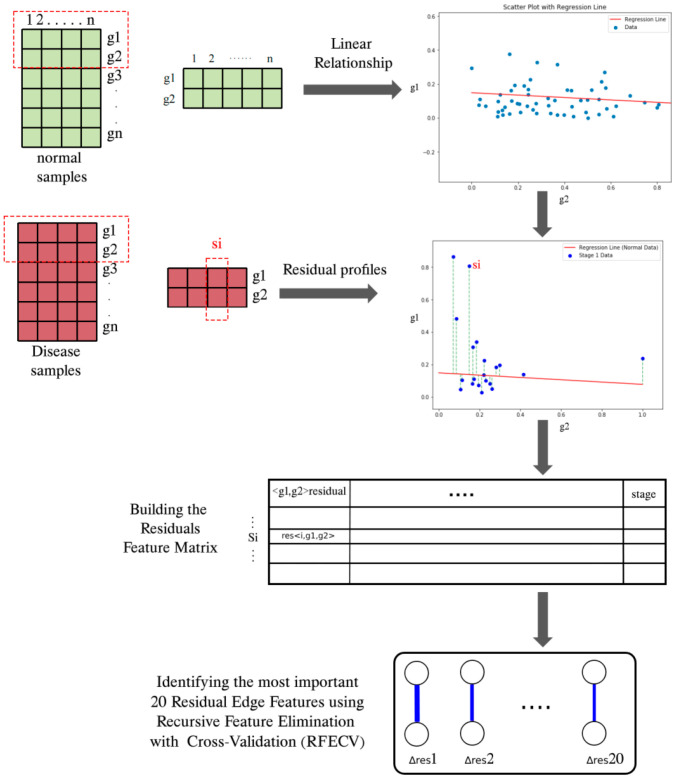
Workflow of identifying the top 20 residual edge features. The number on the top red box is the samle IDs, and the symbol on the right are the gene IDs.

**Figure 3 genes-16-00261-f003:**
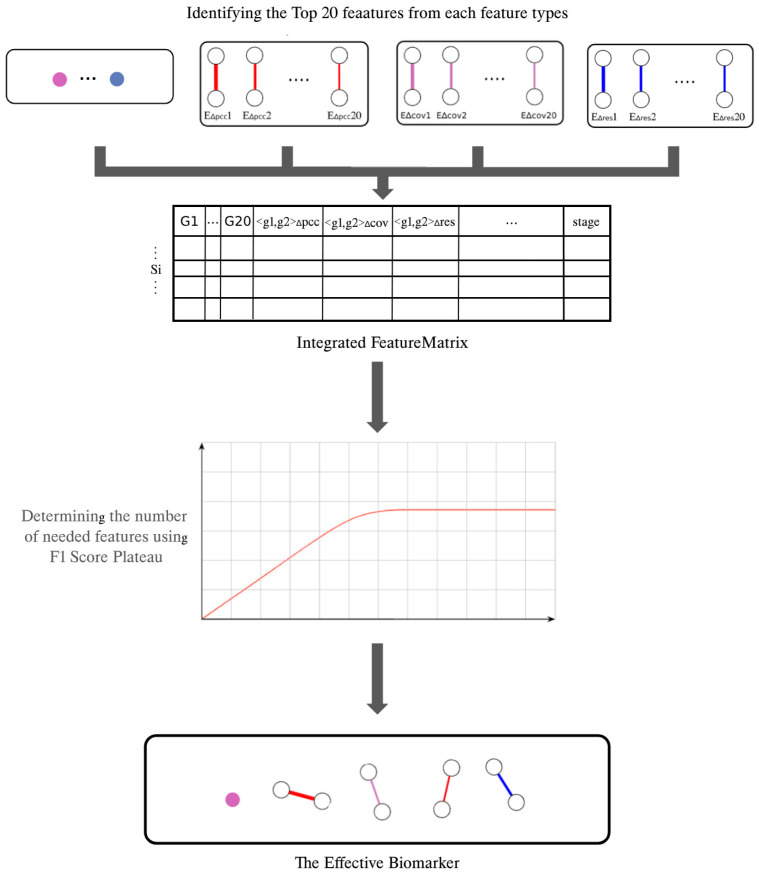
Identifying the most important feature combination set for cancer stage classification.

**Figure 4 genes-16-00261-f004:**
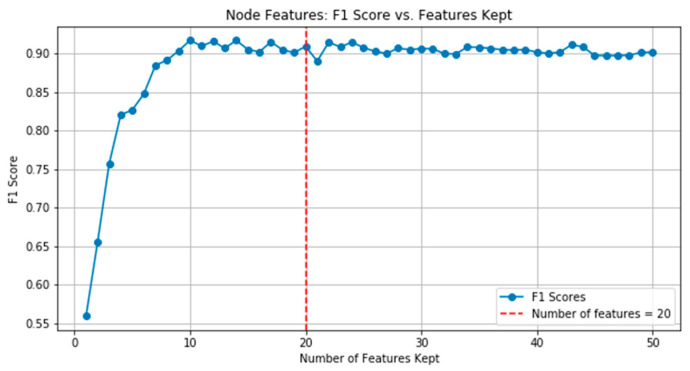
F1 scores of random forest classifiers vs. number of kept node features.

**Figure 5 genes-16-00261-f005:**
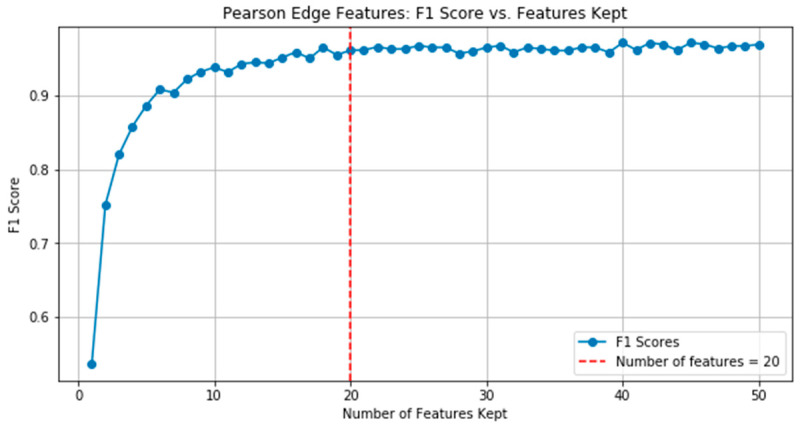
F1 scores of random forest classifiers vs. number of kept person correlation edge features.

**Figure 6 genes-16-00261-f006:**
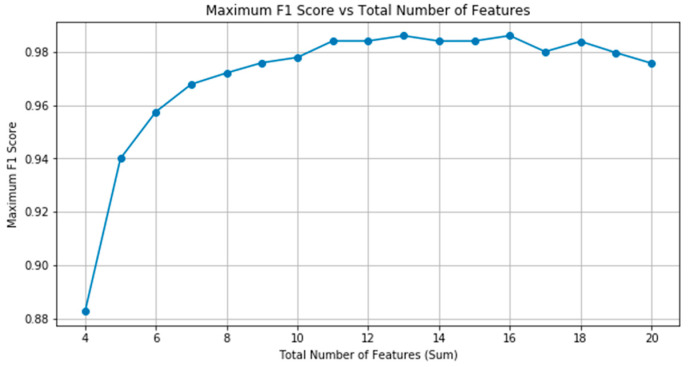
Average F1 score vs. the total number of features.

**Figure 7 genes-16-00261-f007:**
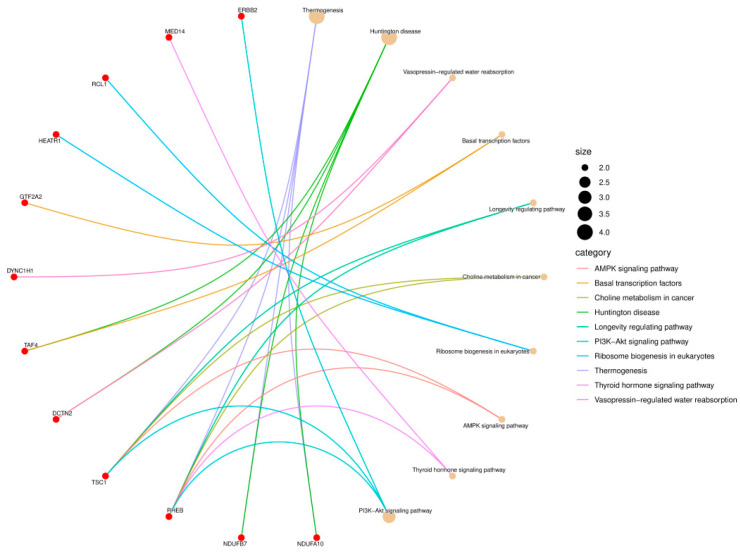
KEGG pathway enrichment result of LUAD.

**Figure 8 genes-16-00261-f008:**
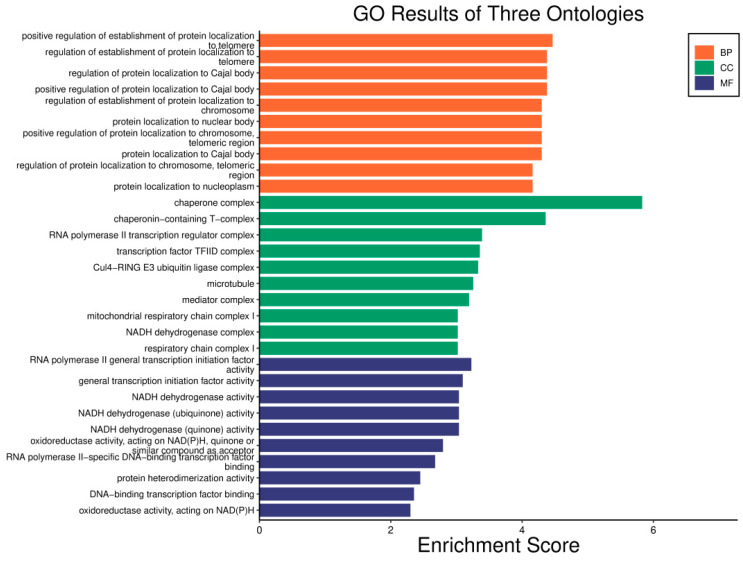
GO enrichment results of stage biomarkers for LUAD.

**Table 1 genes-16-00261-t001:** Top 20 selected node features.

*C20orf160*	*SEMA3G*	*CLEC3B*	*ACVRL1*	*C17orf53*
*PAICS*	*DENND3*	*SGEF*	*CDC6*	*RTKN2*
*PPAT*	*SPAG5*	*ORC6L*	*LMO2*	*LRRC32*
*DLGAP5*	*LYVE1*	*HSPA12B*	*GINS1*	*GRK5*

**Table 2 genes-16-00261-t002:** Top 20 selected Pearson correlation features.

*GTF2A2*:*TAF4*	*CD44*:*ERBB2*	*FIP1L1*:*SYMPK*	*DCTN2*:*DYNC1H1*
*INO80*:*NFRKB*	*RHEB*:*TSC1*	*NCOA1*:*PPARG*	*ARHGEF7*:*CDC42*
*CCT2*:*CCT6A*	*MED1*:*MED27*	*SNAP29*:*STX16*	*SMAD3*:*TGFBR1*
*CDK6*:*CDKN2C*	*RHEB*:*RPTOR*	*LAT*:*PLCG1*	*NDUFA5*:*NDUFS5*
*PRKDC*:*XRCC4*	*ASH2L*:*MEN1*	*TLN1*:*VCL*	*TGFBR1*:*ZFYVE9*

**Table 3 genes-16-00261-t003:** Top 20 selected covariance features.

*CCT2*:*CCT6A*	*MED14*:*MED29*	*DCTN2*:*DYNC1H1*	*TAF6*:*TRRAP*
*ESR1*:*HDAC4*	*PRKDC*:*XRCC4*	*FAM21C*:*KIAA0196*	*EIF4A3*:*RBM8A*
*MED1*:*MED29*	*AEBP2*:*JARID2*	*SMAD3*:*TGFBR1*	*LAT*:*PLCG1*
*MED1*:*MED27*	*AKT1S1*:*RPTOR*	*AMBRA1*:*PIK3C3*	*TLN1*:*VCL*
*EGFR*:*ERBB2*	*FIP1L1*:*SYMPK*	*CDK6*:*CDKN2C*	*HEATR1*:*RCL1*

**Table 4 genes-16-00261-t004:** Top 20 selected residual features.

*NDUFA10*:*NDUFB7*	*ARHGEF7*:*NCK2*	*CUL4A*:*RBX1*	*CUL4A*:*DDB1*
*NDUFA10*:*NDUFA2*	*HDAC4*:*YWHAE*	*CUL4A*:*DDA1*	*ACAD9*:*ECSIT*
*NDUFA10*:*NDUFB3*	*RICTOR*:*RPTOR*	*CCT2*:*CCT6A*	*RB1*:*TFDP1*
*NDUFA10*:*NDUFS7*	*CUL4A*:*DCAF15*	*CUL4A*:*DCAF4*	*E2F2*:*TFDP1*
*ACTL6A*:*INO80*	*ARHGEF7*:*PAK1*	*NCOA1*:*PPARG*	*RHEB*:*TSC1*

**Table 5 genes-16-00261-t005:** Composition of optimal feature combination for lung adenocarcinoma stage classification.

Feature Type	Count	Features
Node Attributes	1	*C20orf160*
Pearson Correlation	4	(*GTF2A2*:*TAF4*), (*RHEB*:*TSC1*), (*CD44*:*ERBB2*), (*CCT2*:*CCT6A*)
Residuals	2	(*NDUFA10*:*NDUFB7*), (*CUL4A*:*DDA1*)
Covariance	4	(*CCT2*:*CCT6A*), (*MED14*:*MED29*), (*DCTN2*:*DYNC1H1*), (*HEATR1*:*RCL1*)

**Table 6 genes-16-00261-t006:** Comparison of F1 scores across feature types and the integrated feature combination.

Feature Type	F1 Score (Best 11 Features)
Node Attributes	0.912
Covariance	0.944
Pearson Correlation	0.951
Residuals	0.953
Integrated Feature (Our Approach)	**0.982**

**Table 7 genes-16-00261-t007:** Comparative performance of biomarker identification methods for LUAD staging.

Methods	Number of Features	F1 Score
WGCNA	151	0.403
ReliefF	100	0.422
SFR-GSN	7	0.968
Integrated Feature (Our Approach)	11	**0.982**

## Data Availability

The original RNA-Seq data presented in the study are openly available in the UCSC Xena repository, with the accession number TCGA Lung Adenocarcinoma (LUAD), and the url is https://tcga-xena-hub.s3.us-east-1.amazonaws.com/download/TCGA.LUAD.sampleMap%2FHiSeqV2.gz (accessed on 1 January 2024); the original PPI data presented in the study are openly available in the STRING repository, accession number Homo sapiens, the url is https://stringdb-downloads.org/download/protein.physical.links.v12.0/9606.protein.physical.links.v12.0.txt.gz (accessed on 1 January 2024). The processed RNA-seq data can be obtained from the Appendix A.

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
