# Peer review of "Identifying Cancer Stage-Related Biomarkers for Lung Adenocarcinoma by Integrating Both Node and Edge Features"

_genes, 2025, doi:10.3390/genes16030261_

Round 1

Reviewer 1 Report

Comments and Suggestions for Authors

The genes feature and their connections are crucial for understanding the molecular basis of diseases and characterize the phenotypes. The existed research method cannot adapt to the subtle changes that often leads to poor accuracy in classifying cancer staging. In the current manuscript, the authors have developed a computational framework to diagnose lung cancer. The proposed method can diagnose each individual test sample, and improve the diagnostic accuracy with minimum feature set. I believe this method is effective in promoting personalized medicine and is potential to improve the clinical outcomes for cancer patients based on the precise diagnosis. This work is very practical in clinic, it worth being known for more readers. Below are my comments to help improve the quality of this manuscript.

  1. Line 42 of “without considering their interactions”. I read the cited literature and found they have considered the interactions in the data processing. Here is the quote from the original paper: “For each species, we downloaded the biomolecular interaction networks from various databases, including BioGrid (http://www.thebiogrid.org), TRED, KEGG (http://www.genome.jp/kegg), and HPRD (http://www.hprd.org)." Can you please explain more on the differences between current available research method and your method? Based on the differences, you can explain that you have improved the accuracy of cancer staging diagnosis. And I believe this is the novelty and the highlight of this study.
  2. Line 42 of “M. Mairé et al. [4] used 54 gene expression signatures as biomarkers for predicting lymph node metastasis”, I found the cited literature does not match with the content here. Can you please correct it? Same mistake as Line 46 of “M. Heck et al. [5] discussed”, and Line 51 of “G. E. Lind et al. [7] studied”, Line 58 of “Lu et al. [15] “ and etc. I can not list them all here, apparently, all the references are messed up. Can you please double check for all the citations and correct them? Besides, the format of references are not consistent, please keep them in the same format.   

Author Response

Comments 1: [The genes feature and their connections are crucial for understanding the molecular basis of diseases and characterize the phenotypes. The existed research method cannot adapt to the subtle changes that often leads to poor accuracy in classifying cancer staging. In the current manuscript, the authors have developed a computational framework to diagnose lung cancer. The proposed method can diagnose each individual test sample, and improve the diagnostic accuracy with minimum feature set. I believe this method is effective in promoting personalized medicine and is potential to improve the clinical outcomes for cancer patients based on the precise diagnosis. This work is very practical in clinic, it worth being known for more readers.]

Response 1: [Thank you very much for those positive comments and suggestion to our manuscript. We will work harder towards to the personalized medicine in the future.]

Comments 2: [Line 42 of “without considering their interactions”. I read the cited literature and found they have considered the interactions in the data processing. Here is the quote from the original paper: “For each species, we downloaded the biomolecular interaction networks from various databases, including BioGrid (http://www.thebiogrid.org), TRED, KEGG (http://www.genome.jp/kegg), and HPRD (http://www.hprd.org)." Can you please explain more on the differences between current available research method and your method? Based on the differences, you can explain that you have improved the accuracy of cancer staging diagnosis. And I believe this is the novelty and the highlight of this study.]

Response 2: [Thank you for pointing this out. As you mentioned in the next comment, we made the citation of reference messed up. And are very sorry to make the confusion for you to understand our idea. In this revision, we have clearly checked all the reference of our manuscript, and hope all the misleading can be corrected. All the updated citations are highlight in red in this revision from line 36 to line 100.]

Comments 3: [Line 42 of “M. Mairé et al. [4] used 54 gene expression signatures as biomarkers for predicting lymph node metastasis”, I found the cited literature does not match with the content here. Can you please correct it? Same mistake as Line 46 of “M. Heck et al. [5] discussed”, and Line 51 of “G. E. Lind et al. [7] studied”, Line 58 of “Lu et al. [15] “ and etc. I can not list them all here, apparently, all the references are messed up. Can you please double check for all the citations and correct them? Besides, the format of references are not consistent, please keep them in the same format.]

Response 3: [Thank you for pointing this out. We are very sad that we did not prepare the reference carefully, which made the citation of references messed up. In this revision, we have clearly checked all the reference of our manuscript, and update them in this revision. All the them are highlight in red in this revision from line 36 to line 100.]

Reviewer 2 Report

Comments and Suggestions for Authors

This study is a bioinformatics exercise to identify gene nodes and edges that are relevant for the staging of lung carcinoma. The text is well written, it is easy to read and understand. The methodology is comprehensible but in the current form it may be difficult to reproduce. The dataset and detailed methodology could be improved for reproducibility.

Additional comments:

(1) Please add the website link and reference for "EdgeR" software.

(2) As I understand, the study used the LUAD-TCGA and GEO datasets. It is important to know the clinicopathological characteristics of the 3 series.

(3) Are the RNA seq different in the TCGA and GEO datasets. How could you reliably merge the gene expression of data from different datasets adn experimental conditions?

(4) Lines 102-103. As I understand, this is the "merged" final dataset. Is the variable "stage" the only one that was evaluated? 

(5) Line 106. Regarding "This dataset includes gene expression profiles of 31823 genes and requires strict quality control measures". 

5.1. Please confirm the final total number of genes (the human reference genome containg around 20,000 protein-coding genes; although numbers are variable).

5.2. Sorry to ask but, what are "quality control measures"?

(6) Line 109. It is stated that some genes were excluded for being "non variant/non informative". Therefore, what is the final dataset? Could you please upload as supplementary data the final dataset that was used in the bioinformatics analysis?

(7) How many cases were included in the training and testing sets?

(8) In section 2.2. How the statistical analyses were performed? What software/programming language/platform were used?

(9) Please confirm that all mathematical formulas are correct.

(10) In section 2.3. Are you using a covariance matrix estimation analysis?

(11) Is the F1 score evaluating the performance of the classification method into the differen stages? 

(12) Section 2 no references at all. Are all methodology of own invention?

(13) When calculating the plots of test accuracy/number of feature, did the accuracy decreased above a determined number of seleted features?

(14) Table 1. Plase write gene names in italics.

(15) What is the relevance of C20orf160 and C17orf53?

(16) In Table 7, do number of "features" equals to number of "genes".

(17) Line 327. What is the function of RHEB and the relevance in lung cancer?

(18) Do the highlighted edges have a prognostic value for the patients? Do they correlate with the survival of the patients?

Author Response

Comments 1: [This study is a bioinformatics exercise to identify gene nodes and edges that are relevant for the staging of lung carcinoma. The text is well written, it is easy to read and understand. The methodology is comprehensible but in the current form it may be difficult to reproduce. The dataset and detailed methodology could be improved for reproducibility.]

Response 1: [Thank you very much for those positive comments and valuable suggestions to our manuscript. We have modified our data available statement and more details of the methods in this revision.]

Comments 2: [Please add the website link and reference for "EdgeR" software.]

Response 2: [Thank you very much for the comment. We have updated the citation of edgeR reference in [21], and have add the website link at line 82 to line 83. All of those changes are highlight in red in this revision.]

Comments 3: [As I understand, the study used the LUAD-TCGA and GEO datasets. It is important to know the clinicopathological characteristics of the 3 series.]

Response 3: [Thank you very much for the comment. It is very important to understand those characteristics. Our method is also proposed for interpret this complex disease and try to find the potential useful edge biomarkers for it. In this revision, we have modified our data availability statement and highlight them in red at line 380 to line 385.]

Comments 4: [Are the RNA seq different in the TCGA and GEO datasets. How could you reliably merge the gene expression of data from different datasets adn experimental conditions?]

Response 4: [Thank you very much for pointing our mistake for citation of GEO database. The right citation should be the UCSC Xena repository. It is an easy-to-use data files derived from public resources like TCGA, so the date is actually the same and does not need to merge. In the revision, we have modified this correctly.]

Comments 5: [Lines 102-103. As I understand, this is the "merged" final dataset. Is the variable "stage" the only one that was evaluated?]

Response 5: [Thank you very much for your comments. As we have clarified in the last response, the RNAseq dataset was obtained from UCSC Xena repository, where the raw data was TCGA LUAD cohort. It does not need to merge two datasets. The samples are grouped according to their pathological stage, which is available according to their sample description file.]

Comments 6: [Line 106. Regarding "This dataset includes gene expression profiles of 31823 genes and requires strict quality control measures".

- Please confirm the final total number of genes (the human reference genome containg around 20,000 protein-coding genes; although numbers are variable).

- Sorry to ask but, what are "quality control measures"?]

Response 6: [Thank you very much for your comments. In this RNA seq data, the expression was given according to individual transcript, and each transcript was given a “gene name”. This is why there are more than 20,000 transcripts are given here. Since they used the similar ID to gene name, this is why we say there are 31823 genes here. To make it more accurate, we use the name of transcript to replace gene to make it more clear and highlight it in red in this revision in line 111. The quality control measure is the descript later, which is “Transcript with zero expression in over 50% of the samples were excluded to eliminate non informative variables and enhance the analytical robustness of the study.”]

Comments 7: [Line 109. It is stated that some genes were excluded for being "non variant/non informative". Therefore, what is the final dataset? Could you please upload as supplementary data the final dataset that was used in the bioinformatics analysis?]

Response 7: [Thank you very much for your comments. The final number of genes remained for analyzing is 13163 in this study. We have updated this information in this revision and make those datasets publicly available as well.]

Comments 8: [How many cases were included in the training and testing sets?]

Response 8: [A 10-fold cross validation method was used in this study, which means there are 90% of samples in the training set and the rest of 10% are in the testing set. We highlighted this information in the revision at line 129 and line 250.]

Comments 9: [In section 2.2. How the statistical analyses were performed? What software/programming language/platform were used?]

Response 9: [The R software and EdgeR package are employed in this study to conduct the statistical analysis, including the data normalization, differently expressed gene identification, their p-values, and the false discovery rate.]

Comments 10: [Please confirm that all mathematical formulas are correct.]

Response 10: [Thank you very much for your comments. We have double checked every formulas.]

Comments 11: [In section 2.3. Are you using a covariance matrix estimation analysis?]

Response 11: [In this section, three methods are employed, which are the change of the correlation coefficients, the change of covariance, and the residual errors. Covariance matrix is one of the method we used in this study.]

Comments 12: [Is the F1 score evaluating the performance of the classification method into the different stages?]

Response 12: [In this study, we would like to classify patients into different stages. So a multiclass classification was conducted using random forest classifier. The F1 score is calculated from this multiclass classification according to the prediction results of all individual samples in the cross validations. All stages are predicted together, and only the final F1 score are outputted eventually. To make it clearer, we have modified our manuscript at line 84 and line 218. Both of them are highlight in red.]

Comments 13: [Section 2 no references at all. Are all methodology of own invention?]

Response 13: [There are three methods introduced in section 2. The method 1 and method 3 were adopted from previous literature and the method 2 are our own invention. To make these more clear, we add those previous literature in this revision at line 150 and line 196.]

Comments 14: [When calculating the plots of test accuracy/number of feature, did the accuracy decreased above a determined number of seleted features?]

Response 14: [Yes. The x-axis is the number of features kept, and the y-axis is the final f1-scores. Since the performance is a combination of the used features, so even the number of feature increase gradually, the eventual f1-scores are fluctuating.]

Comments 15: [Table 1. Plase write gene names in italics.]

Response 15: [Thank you very much for your comments. We have modified accordingly in this revision.]

Comments 16: [What is the relevance of C20orf160 and C17orf53?]

Response 16: [Thank you very much for your comments. There is no direct relevance between C20orf160 and C17orf53. Both of them were selected as node features based on the random forest classifier. They are good at distinguish samples from individual stages. Those node features are used as a comparison to the following edge features. We argue that the obtained edges features are not only achieve higher accuracy, but also need smaller number of features, which is also the main contribution of the proposed method.]

Comments 17: [In Table 7, do number of "features" equals to number of "genes".]

Response 17: [Thank you very much for your comments. It depends on what kinds of features are employed. If the node features are used, then the number of features equals to the number of genes. But if the edge features are used, then the number of features equals to the number of gene-pairs.]

Comments 18: [Line 327. What is the function of RHEB and the relevance in lung cancer?]

Response 18: [Thank you very much for your comments. The RHEB gene is the one we identified from the proposed method. This gene involves in six pathways, which has the most number of connections. It was related to lung cancer, where the reference are as follows:

[1] Liu X, Wang X, Chai B, Wu Z, Gu Z, Zou H, Zhang H, Li Y, Sun Q, Fang W, Ma Z. miR-199a-3p/5p regulate tumorgenesis via targeting Rheb in non-small cell lung cancer. Int J Biol Sci. 2022, 18(10): 4187-4202.

[2] Zheng H, Liu A, Liu B, Li M, Yu H, Luo X. Ras homologue enriched in brain is a critical target of farnesyltransferase inhibitors in non-small cell lung cancer cells. Cancer Lett. 2010;297:117-125.

We have added those reference in this revision to make it clearer in line 333.]

Comments 19: [Do the highlighted edges have a prognostic value for the patients? Do they correlate with the survival of the patients?]

Response 19: [Thank you very much for your comments. In this study, we have not investigated in this area. We think your comments is very valuable, and we will investigate such correlation in our future studies.]